# A study protocol for the assessment of antibiotic use and bacterial antimicrobial resistance among children under five years of age: Implications for a resource-limited setting

Onome T. Abiri[1]*, Troy D. Moon[2], John S. Schieffelin[3], Cynthia A. Adinortey[4], Gustavo Amorim[5], Isatta Wurie[6], Donald S. Grant[7], Babatunde Duduyemi[8], Mohamed Samai[1]

**1** Department of Pharmacology and Therapeutics, Faculty of Basic Medical Sciences, College of Medicine and Allied Health Sciences, University of Sierra Leone, Freetown, Sierra Leone, **2** Department of Tropical Medicine and Infectious Diseases, Tulane University Celia Scott Weatherhead School of Public Health and Tropical Medicine, New Orleans, Louisiana, United States of America, **3** Sections of Paediatric and Adult Infectious Diseases, Tulane University School of Medicine, New Orleans, Louisiana, United States of America, **4** Department of Molecular Biology and Biotechnology, University of Cape Coast, Cape Coast, Ghana, **5** Department of Biostatistics, Vanderbilt University Medical Centre, Nashville, Tennessee, United States of America, **6** Department of Chemical Pathology, Faculty of Laboratory Medicine and Diagnostics, College of Medicine and Allied Health Sciences, University of Sierra Leone, Freetown, Sierra Leone, **7** Department of Community Medicine, Faculty of Clinical Sciences, College of Medicine and Allied Health Sciences, University of Sierra Leone, Freetown, Sierra Leone, **8** Department of Pathology, Faculty of Basic Medical Sciences, College of Medicine and Allied Health Sciences, University of Sierra Leone, Freetown, Sierra Leone

* berylonome@gmail.com

## Abstract

### Background

The irrational use of antibiotics to treat infections in children is a crucial contributing factor to bacterial antimicrobial resistance (AMR), which can have economic and health consequences, such as morbidity and mortality. This study aims to evaluate antibiotic use and AMR in children under five years of age in Sierra Leone.

### Methods

This study will be conducted in three hospitals: Ola During Children, Kenema Government, and Magburaka Government Hospitals in Sierra Leone, among healthcare professionals and patients. A mixed-method (qualitative and quantitative) approach will be used to evaluate paediatric health professionals' knowledge, perceptions, and antibiotic prescription practices. Additionally, two cross-sectional sub-studies will assess inpatient and outpatient trends in antibiotic use and consumption in children, and a cross-sectional observational sub-study will investigate bacterial profiles and AMR among children with bloodstream infections. The anatomical therapeutic chemical (ATC) and the World Health Organisation Access, Watch and Reserve (WHO

**Data availability statement:** The protocol and all deidentified data generated will be made publicly available when the study is completed and published.

**Funding:** This research is supported by the Fogarty International Centre of the National Institutes of Health under Award Number D43TW011248. The content is solely the responsibility of the authors and does not necessarily represent the official views of the National Institutes of Health. The funders had no role in study design, data collection and analysis, decision to publish, or preparation of the manuscript.

**Competing interests:** The authors declare that they have no competing interests.

**Abbreviations:** AMR: Bacterial Antimicrobial Resistance; ARIMA: Autoregressive Integrated Moving Average; AST: Antibiotic Susceptibility Test;ATC: Anatomic Therapeutic Chemical Classification; AWaRe: Access Watch and Reserve; BSI: Bloodstream infections; DDD/1000PDs: Defined Daily Dose/1000 patient days; DDD/100BDs: Defined Daily Dose/100 bed days; DOT/1000PDs: Days of Therapy/1000 patient days; DOT/100BDs: Days of Therapy/100 bed days; ESBL: Extended Spectrum Beta-Lactamase; ETAT＋: Emergency Triage Assessment and Treatment plus; EUCAST: European Union Committee on Antibiotic Susceptibility Testing; KAP: Knowledge, Attitude, and Practice; KGH: Kenema Government Hospital; MIC: Minimum Inhibitory Concentration; MGH: Magburaka Government Hospital; MRSA: Methicillin-Resistant Staphylococcus aureus; ODCH: Ola During Children's Hospital; SLESRC: Sierra Leone Ethics and Scientific Review Committee; TAPBM: Teixeria Antibiotic Prescribing Behavioural Model; WAR: Western Area Rural; WAU: Western Area Urban; WHO: World Health Organisation.

AWaRe) classifications, days of therapy per 1,000 patient days (DOT/1000PDs), and days of therapy per 100 bed days (DOT/100BDs) will be used to determine the use and consumption. The DOT/1,000PDs and DOT/100BDs will be compared with the defined daily dose/1,000 patient days (DDD/1000PDs) and defined daily dose/100 bed days (DDD/100BDs), respectively. A pre-tested interview guide, interviewer-administered questionnaire and data collection tools adapted from previous studies will be employed for data collection. The sample sizes will be determined, and appropriate sampling methods will be used. Data will be analysed thematically using NVivo 15, and descriptive and inferential statistics using the R software.

## Discussion

The results of this study will inform policymakers and healthcare professionals in developing and/or implementing policies, guidelines, and educational initiatives that will promote antibiotic stewardship among children in Sierra Leone.

## Background

Sierra Leone faces significant child health and survival challenges, with one of the world's highest under-five mortality rates of approximately 122 deaths per 1,000 live births in 2019 [1,2]. This high mortality rate is due to inadequate healthcare resources, poor diagnostic facilities, socioeconomic factors, and the high burden of infectious diseases [3,4]. The high prevalence of infectious diseases among children under five years of age is particularly concerning because their developing immune systems make them especially vulnerable, often leading to frequent antibiotic prescriptions [5,6]. Increased exposure to infectious agents and improper medication use may accelerate the development of bacterial antimicrobial resistance (AMR), threatening effective treatment and child health [5,7]. Additionally, healthcare professionals' inconsistent knowledge, attitudes, and practices, along with poor adherence to clinical guidelines, contribute to inappropriate prescribing habits and the emergence of resistance [8–10].

To address the urgency of combating AMR in Sierra Leone, the Ministries of Health, Agriculture, Forestry and Food Security, the Environment Protection Agency, and other stakeholders carried out a preliminary country situational analysis in 2017. The results informed the development of the Sierra Leone AMR strategic plan (2018–2022), aimed at guiding government and partner activities in combating AMR in Sierra Leone [11].

Studies on the knowledge, attitudes, and prescribing practices of paediatric health professionals have yielded mixed results, indicating both poor and good knowledge, positive and negative attitudes, and varied antibiotic prescription practices. Physicians demonstrated good knowledge and positive attitudes regarding appropriate use; however, inappropriate prescribing practices were still observed, highlighting a gap between knowledge and practice [12–15]. Additionally, external factors such as

time pressure, parental demand for antibiotics, and financial incentives from the pharmaceutical industry have been cited as influencing prescribing practices [16,17].

Understanding antibiotic use patterns in outpatient and inpatient settings is critical. Pharmacoepidemiologic methods such as the Anatomic Therapeutic Chemical Classification (ATC), Days of Therapy/1000 patient days (DOT/1000PDs), and DOT/100 bed days (DOT/100BDs) are commonly used to analyse data, provide insights into paediatric antibiotic use, identify trends, and suggest areas for policy and practice improvement. Globally, studies have reported high antibiotic prevalence (7–80%) and consumption (400–800 DOTs/1000PDs) in outpatient settings, irrespective of economic development [18–20]. In hospitalised children, antibiotic prevalence rates vary (20–98%) and consumption ranges (50–100 DOT/100 BDs), with calls for better stewardship and guideline adherence [21–26]. Studies have also linked antibiotic use to demographic and clinical factors, such as age, length of hospital stay, ward type, comorbidities, and clinical indications [27,28].

Furthermore, studies have documented a high prevalence of bloodstream infections (10–15%) among children in sub-Saharan Africa, South America, and Southeast Asia, along with significant rates of AMR in bacterial isolates, including methicillin-resistant *Staphylococcus aureus* (MRSA) and extended-spectrum beta-lactamase (ESBL)-producing Enterobacteriaceae, such as *Klebsiella pneumoniae and Escherichia coli* [29–31].

Given the lack of data on antibiotic use and AMR among children in Sierra Leone, studies are essential for generating local data. Identifying the gaps in stewardship and demographic factors influencing antibiotic use will aid in developing evidence-based interventions and policies to promote rational use and curb the spread of AMR. These data will support both local and global efforts to combat AMR and to preserve antibiotic effectiveness in the future. Research on antibiotic use and resistance among children in Sierra Leone is crucial for improving healthcare outcomes and protecting public health in this vulnerable group.

This study will investigate antibiotic use and AMR among children under five years of age in Sierra Leone. The specific objectives are:

1. To determine the knowledge and attitude, and explore factors influencing antibiotic prescribing practices regarding antibiotic use and AMR among paediatric healthcare professionals in three Sierra Leonean hospitals.

2. To prospectively evaluate outpatient antibiotic use among children in these hospitals using the ATC and WHO AWaRe classifications, and DDD/1,000PDs, and DOT/1,000PDs.

3. To assess inpatient antibiotic use and consumption patterns in three hospitals using the ATC and WHO AWaRe classifications, DDD/100BDs and DOT/100BDs, with two sub-objectives:

   3.1   To prospectively evaluate inpatient antibiotic use over three months.

   3.2   To retrospectively analyse trends in inpatient antibiotic use from 2019-2023.

4. To investigate bacterial profiles, AMR, and associated factors in children with bloodstream infections (BSI) at a paediatric tertiary referral hospital in Sierra Leone.

We hypothesise that paediatric healthcare professionals in Sierra Leone will demonstrate strong knowledge and positive attitudes about antibiotic use and the necessity to reduce AMR. Nonetheless, antibiotic prescribing practices will remain high in both inpatient and outpatient settings, even when antibiotics may not be necessary. Additionally, there will be a high prevalence of AMR among bacterial pathogens causing BSIs in children. Investigating the determinants of these high prescribing practices will identify areas for intervention to reduce unnecessary antibiotic use and curb AMR development.

## Materials and methods

### Study setting

Sierra Leone has five administrative regions and 16 districts: Eastern, Northern, Southern, Northwestern Provinces, and Western Area. The Western Area has two districts: Western Area Urban (WAU), encompassing the capital, Freetown,

and Western Area Rural (WAR). The Eastern and Southern provinces have three (Kenema, Kono, and Kailahun) and four districts (Bo, Pujehun, Moyamba and Bonthe) with Kenema and Bo as the regional district headquarters respectively. The Northern and North-West provinces have four (Bombali, Tonkolili, Koinadugu and Falaba) and three districts (Portloko, Kambia and Karene) with Bombali and Portloko as the regional district headquarters. The health system is structured into three tiers: primary, secondary, and tertiary care. This study will be conducted at Freetown's only teaching tertiary paediatric hospital (Ola During Children), a regional secondary healthcare facility in Kenema, and a district secondary hospital in Magburaka, Tonkolili (Fig 1). These hospitals provide both inpatient and outpatient paediatric services. Data collection will commence on 1st March 2025, will be completed in February 2026. Overall study period is 2024–2026.

## Study design and research framework

Objective one will employ a sequential mixed-method approach (qualitative and quantitative) to assess paediatric health professionals' knowledge, perceptions, and antibiotic prescribing practices. Objectives two and three involve two separate cross-sectional studies to evaluate outpatient and inpatient antibiotic use and consumption respectively. Objective three is divided into sub-objectives 3.1 and 3.2, with 3.1 being a prospective cross-sectional study and 3.2 a retrospective inpatient trend analysis. Finally, objective four will be a cross-sectional prospective observational study examining bacterial profiles and AMR in hospitalised children with BSI.

This research will utilise a hybrid theoretical-conceptual framework adapted from the Teixeira Antibiotic Prescribing Behavioural Model (TAPBM) (Fig 2). This framework identifies intrinsic and extrinsic predictors of antibiotic prescriptions based on the theory of knowledge, attitudes, and practices (KAP) and the examination of external determinants influencing prescribing practices [17]. The conceptual component employed a model from previous studies [30,32]. The

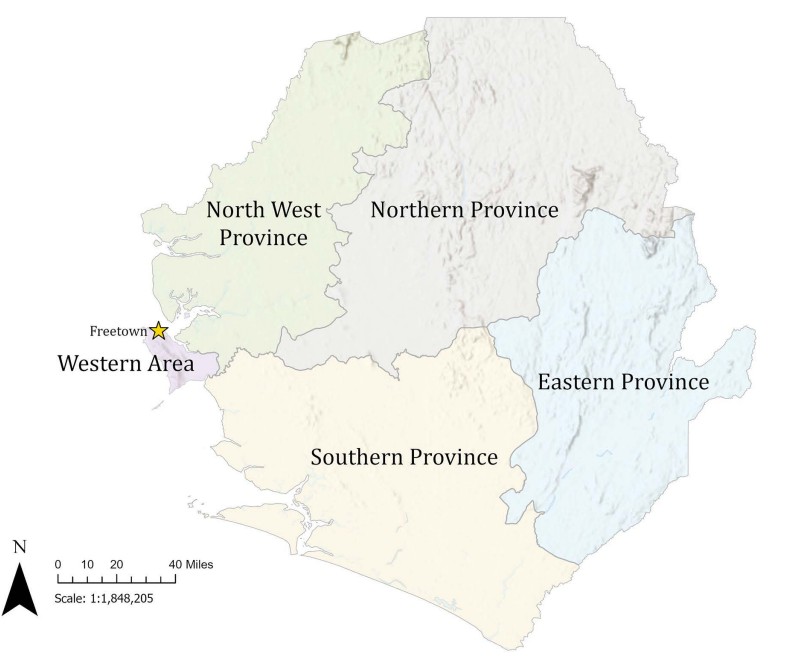

**Fig 1. Administrative map of Sierra Leone showing the locations of the three study sites (created by Katherine A. Gruber).** The map was created entirely with our own dataset on top of an ESRI base map. No ESRI sample data or other third-party datasets were used. *Sources: Esri, TomTom, Garmin, FAO, NOAA, USGS, © OpenStreetMap contributors, and the GIS User Community, ESRI, USGS.* ODCH = Ola During Children Hospital in Freetown, KGH = Kenema Government Hospital, in Kenema District, MGH = Magburaka Government Hospital in Tonkolili District.

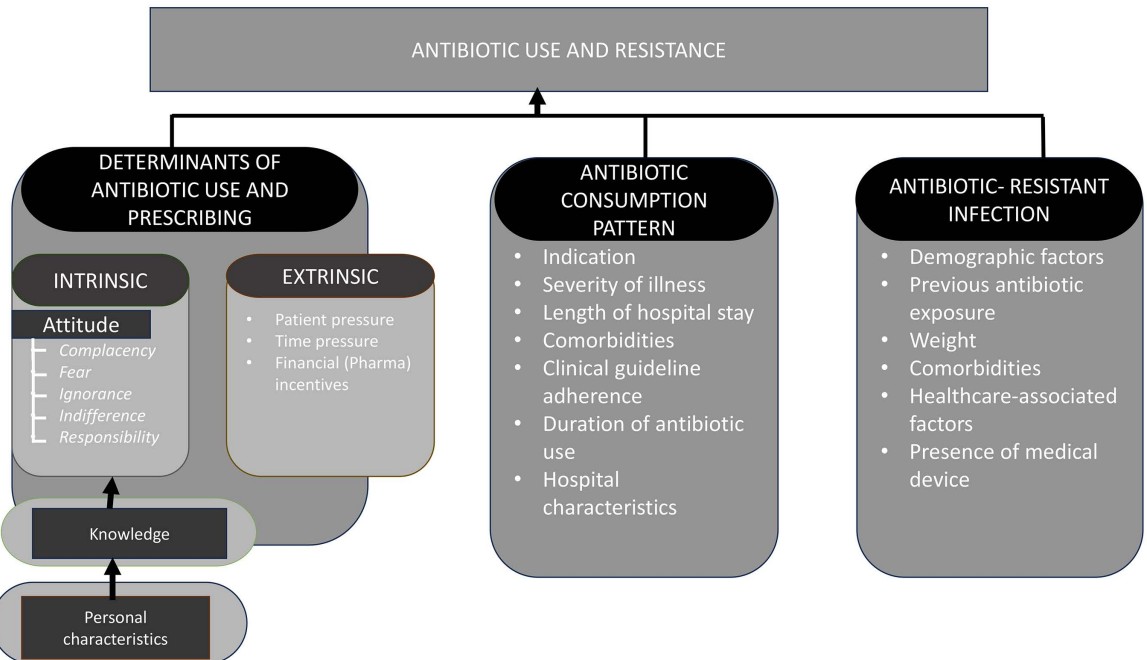

**Fig 2. Hybridized theoretical-conceptual framework for describing predictors of antibiotic use and resistance.**

framework is divided into three broad domains to guide our exploration of antibiotic use and resistance in Sierra Leone: 1) Determinants of antibiotic use and prescribing, divided into a) intrinsic and b) extrinsic factors; 2) Antibiotic consumption patterns; and 3) Antibiotic-resistant infections.

## Operational definitions

*DOT method:* To estimate antibiotic use using the DOT method, one DOT will represent the administration of a single agent on a given day, regardless of the number of doses administered or dosage strength. To express aggregate use, total DOTs will be standardised to 1000 patient days (DOT/1,000PDs) and 100 bed days (DOT/100BDs) for outpatient and inpatient settings respectively to allow comparison between services of different sizes [33,34].

*DDD method:* DDD is the assumed average maintenance dose per day for a drug used for its main indication in adults, as specified by the World Health Organization (WHO). The DDDs will be calculated by converting the total antibiotics dispensed into grammes and dividing by the WHO-assigned DDDs based on the 2023 ATC/DDD index. To express aggregate use, total DDDs will be standardised to 1000 patient days (DDD/1,000PDs) and 100 bed days (DDD/100BDs) for outpatient and inpatient settings respectively to allow comparison between services of different sizes [33–35].

## Specific objective 1

To determine the knowledge and attitude, and explore factors influencing antibiotic prescribing practices regarding antibiotic use and AMR among paediatric healthcare professionals in three Sierra Leonean hospitals.

**Study population.** This will include all paediatric healthcare prescribers (nurses, medical doctors, and community health officers) at the study sites. Consenting participants working at the study sites will be invited to participate.

**Sample size and sampling method.** For the qualitative study, purposive and snowball sampling will be used to recruit participants for the qualitative part until full data saturation is achieved. We anticipate that 20 in-depth interviews will be

conducted. The quantitative study will include all paediatric healthcare prescribers working at the study sites. This will include 23 medical doctors, 131 nurses, and 12 community health officers, giving a total of 166 healthcare professionals.

**Data collection instrument and procedure.** For the qualitative study, semi-structured interviews will be conducted utilizing an interview guide that will be adapted from prior research to examine prescribing practices in children [36–38]. Data will be collected through note taking and each interview will be audio-recorded. Pilot interviews will be conducted to pretest the guide for validity and clarity. The quantitative study will use a structured interviewer-based questionnaire developed from previous studies [38,39]. The questionnaire will be in English and comprise four sections. Section A will gather sociodemographic and professional details, such as age, gender, years of practice, hospital type, and qualifications. Section B will have 18 items assessing knowledge of antibiotic use and resistance using three response options (yes, no, do not know). Correct answers will score 1 point, and incorrect/unsure answers will score 0 points. Section C will include twenty-seven items designed to assess five aspects of healthcare professional's attitudes toward antibiotic prescribing per the Teixeira Antibiotic Prescribing Behavioural Model [40]. These include 1) Ignorance: Lack of concern for antibiotic resistance due to over-prescribing; 2) Responsibility of others: Belief that others are responsible for antibiotic resistance; 3) Indifference: Lack of motivation to change prescribing practices; 4) Complacency: Prescribing to satisfy patient demands; and 5) Fear: Prescribing owing to fear of losing patients or disputes.

Attitude questions will use a five-point Likert scale. For ignorance, responsibility of others, and indifference, options range from "strongly disagree" to "strongly agree." For complacency and fear, options range from "never" to "always." Each aspect's mean score will range from 1 to 5, with higher scores indicating more inappropriate prescribing attitudes. The questionnaire will be pretested for cultural appropriateness and to identify issues with wording, layout, and under-standing. Data will be collected through face-to-face interviews by the investigator and trained research enumerators. Training will cover the study objectives, proposal, interview guide, ethical considerations, and data collection procedures. Participants recruitment and data collection started in March 2025 and will be completed by August 2025. The result is expected by December, 2025.

**Study variables/outcome measures.** The dependent variables are knowledge and attitude. The independent variable will be sociodemographic characteristics (age, sex, qualification, cadre of health professionals, number of years of practice, and type of hospital).

**Data analysis plan.** Inductive and deductive thematic analyses will be used to analyse and derive themes from qualitative data. Audio recordings will be transcribed verbatim and translated into English. The data will be coded, reduced, and organised to develop final themes and subthemes using NVivo version 15. Emerging themes and conclusions will be discussed in regular research meetings to ensure alignment with the study's objectives.

For the quantitative study, data will be collected in a data collection tool encrypted in Epicollect5 version 11 data gathering platform and analysed using R version 4.3.2. Demographic variables and responses to knowledge and attitude statements will be analysed using descriptive statistics, such as frequency and percentage. Overall knowledge and attitude scores will be calculated by dividing the sum of correct answers by the total number of questions. Ordinal regression analysis will identify demographic and professional factors associated with knowledge and attitude. Spearman's rank order correlation coefficient will describe the strength and direction of relationships between knowledge and attitude responses. The significance level for all statistical tests will be set at 0.05 for all statistical tests.

## Specific objective 2

To evaluate outpatient antibiotic utilisation patterns in three hospitals in Sierra Leone.

**Study population.** The study will be conducted prospectively and will include children under five years of age brought to the hospital for outpatient consultation during the study period. Patients scheduled for admission and those with incomplete records will be excluded.

**Sample size determination and sampling.** The sample size n will be determined as the minimum number of participants to estimate the prevalence p of antibiotic use in the outpatient setting in Freetown, Sierra Leone, with a margin of error equal to 2.3%. We used an estimated prevalence of 32.2% [41] and computed the minimum sample size using equation (1) below.

$$n = \frac{Z^2 \times p \times (1-p)}{d^2}$$

(1)

where z is the standard normal deviate of 1.96 which corresponds to 95% confidence level. Using a significance level (i.e., $\alpha = 5\%$), we will need a minimum of 1,585 participants to estimate a prevalence of 32.2%. Considering methodological contingencies, the actual sample size for this study will be 1,762 children after allowing for 10% of missing data. Based on probability proportional to size sampling, the expected sample sizes for the individual hospitals will be 1,026, 421, and 317 for ODCH, KGH and MGH, respectively. Consecutive sampling will be applied to obtain sampling units at the study sites. This sample size will be enough to estimate the underlying prevalence of antibiotic in the population of interest with a margin of error of 2.3%, considering a 95% confidence interval. In addition, it will have at least 90% power to detect differences of at least 0.164 standard deviations in continuous variables, such as the age of the children.

**Data collection instrument and procedure.** The data collection tool for this study will be adapted from previous research and validated prior to use [18,27,41]. The principal investigator and trained data collectors will gather data daily from patient case files to identify prescription patterns, including demographic and clinical information such as age, gender, weight, and hospital type and location. Antibiotic use data, including the amount prescribed and dispensed over three months, will be extracted from prescription and pharmacy records. This includes details on antibiotics prescribed, number of antibiotics per patient prescription, prescriber information, duration of use, and indication. Additional details on the dosage regimen, unit strength, route of administration, and primary diagnosis will also be collected. The investigator, along with infectious disease experts, paediatricians, and a clinical pharmacologist, will review the collected data. The appropriateness of antibiotic prescribing will be evaluated by the investigator and a panel of experts, including paediatricians, infectious disease physicians, pharmacists, nurses, and clinical pharmacologists. This will be done based on the Emergency Triage Assessment and Treatment Plus (ETAT+) guideline and a structured algorithm into appropriate, inappropriate, and not assessable as described by McMullan et al. [42,43]. Participants recruitment and data collection started in April 2025 and will be completed by September 2025. The result is expected by January 2026.

**Study variables/outcome measures.** The primary outcome will be antibiotic prevalence rate according to the ATC and WHO AWaRe classifications, DOT/1000PDs, DDD/1000PDs, and appropriateness of antibiotic prescribing. The independent variables are age, gender, level of hospital, geographical location of the hospital, number of antibiotics per patient, duration of antibiotic use, and indication.

**Data analysis plan.** Data will be entered in a data collection tool encrypted in Epicollect5 version 11 data gathering platform and analysed using R version 4.3.2. Descriptive statistics, such as frequencies and percentages, mean or median will be used to present the demographic and clinical characteristics and treatment-related information of study participants and antibiotic consumption. The prevalence of systemic antibiotic use will be expressed as a percentage of the total patients on systemic antibiotics at the study time against the number of outpatients, categorised according to the WHO Access, Watch, and Reserve (AWaRe) classification [44]. Additionally, antibiotics will be categorised using the ATC Classification System under the J01 category for systemic use [35]. Antibiotic quantities in children will be measured using DOT and DDD calculated from prescriptions, adjusted for the paediatric population in the geographic area, and expressed as DOT/1,000 PDs and DDD/1,000PDs. The DOTs and DDDs will be summed by antibiotic type, stratified by sex, age, hospital, and antibiotic type, and compared. To compare DOT vs. DDD values, an independent t-test will be

used. Univariable and multivariable logistic regression models will be used to analyse associations between dependent and independent variables, with a significance level set at 5%.

## Specific objective 3

To determine antibiotic use and consumption among hospitalised children in three hospitals in Sierra Leone.

**Study population.** Subobjectives 3.1 and 3.2 will be for about three months cover during the study period and five years (2019–2023) respectively. The study will include children under five years of age who were admitted during these periods. Exclusions will apply to patients with single-day admissions for diagnostic tests, chemotherapy, or blood transfusions.

**Sample size determination and sampling.** *For sub-objective 3.1*, the sample size will be obtained using equation (1), using an estimated prevalence of antibiotic use in the inpatient setting equal to 50%. We assumed a margin of error equal to 3% and a 5% significance level, leading to a minimum of 1,067 participants. As, before, considering methodological contingencies, the actual sample size for this study will be 1,186 after allowing for 10% of missingness. Based on probability proportional to size sampling, the sample sizes for the individual hospitals will be 691, 283, and 214 for ODCH, KGH and MGH, respectively. Consecutive sampling will be applied to obtain sampling units at the study sites.

*For sub-objective 3.2,* using again equation (1), with a prevalence of 50% and a margin of error of 2.5%, the estimated sample size is 1,537 participants. Allowing for 10% of missingness, we should recruit 1,708 patients. Based on probability proportional to size sampling, the sample sizes for the individual hospitals will be 994, 408, and 306 for ODCH, KGH and MGH, respectively. Systematic random sampling will be applied to obtain sampling units at the study sites where the selection interval $k = N/n$, where N (the population of inpatients for the three hospitals from 2019-2023 = 87,466 and $n = 1,708$). Therefore $k = 51.20 = 51$. The first sampling number is $S = R*k$, where R (a random number = 0.96 and $k = 52$. Therefore S= 48.96 = 49.

**Data collection instrument and procedure.** The study will adapt and validate data collection tools from previous research [23,24]. Data will be gathered from patient charts, including demographic and clinical details, such as age, sex, hospital type, size, location, number of beds, occupancy rate, length of stay, paediatrician-to-bed ratio, antibiotic class, number of antibiotics per patient, duration of use, indication, and presence of comorbidities. For patients with antibiotic prescriptions, the data will include the name, number, strength, pack size, quantity, administration route, indication, start date, frequency, missed doses, duration, and any changes during the infection episode. The investigator and trained data collectors will collect the data, and the data will be reviewed by the investigator, paediatricians, and infectious disease experts. The appropriateness of antibiotic prescribing will be evaluated as described in specific objective two. Regarding sub-objective 3.1, participants recruitment and data collection commenced in May 2025 and will be completed by October 2025. The result is expected by February 2026. For sub-objective 3.2, recording screening will start in September 2025 and data extractive will start in October 2025. The result is expected in March 2026.

**Study variables/outcome measures.** The dependent variables are inpatient antibiotic prevalence rate according to ATC and WHO AWaRe classifications, DOT/100BDs, DDD/100BDs and appropriateness. The independent variable will be age, gender, level of hospital, size of the hospital, geographical location of the hospital, ward type, occupancy rate, rate of broad-spectrum antibiotics, number of antibiotics per patient, year, length of stay, duration of antibiotic use, indication, and presence of comorbidity.

**Data analysis plan.** Data will be entered into Epicollect5 version 11 and analysed using R version 4.3.2. Descriptive statistics, including mean, standard deviation, median, interquartile range, frequencies, and percentages, will describe categorical and continuous variables, such as demographic and clinical characteristics and antibiotic consumption. The prevalence of systemic antibiotic use will be expressed as a percentage of the total number of patients on systemic antibiotics at the time of the study against the total admissions, categorised according to the WHO AWaRe and ATC classification systems.

Antibiotic consumption in children will be measured using DOT and DDD adjusted for the paediatric population in the geographic area and expressed as DOT/100 BDs and DDD/100BDs which will be summed up by antibiotic type, stratified

by sex, age and hospital. The DOT/100BDs will be compared with the overall DDD/100BDs for each hospital using an independent t-test. Trends in DOT/100BDs and DDD/100BDs (total, per ATC group, oral, intravenous) from 2019–2023 will be analysed using Autoregressive Integrated Moving Average (ARIMA). Subanalyses will consider antibiotic prescription patterns before and after the national COVID-19 containment measures. Univariable and multivariable logistic regression models will be used to determine the associations between dependent and independent variables, with a significance level set at 5%.

### Specific objective 4

To determine the bacterial profile, antibiotic resistance, and associated factors among children with bloodstream infection (BSI) at a paediatric tertiary referral hospital in Sierra Leone.

**Study population.** This study will include all children, regardless of gender, with clinical presentations of BSI who visit the outpatient department or are admitted to the Ola During Children Hospital during the study period [45]. Participation requires voluntary consent from the children's parents or guardians. Enrolment criteria, based on previous studies, will include temperature (>38 °C or <36 °C), fever onset, age-specific tachycardia, age-specific tachypnoea, convulsions, altered consciousness, and abnormal feeding [45,46].

**Sample size determination and sampling.** Sample size will be calculated using equation 1, with p = 50% (estimated prevalence of BSI among children under five years), and a margin of error = 7.5%. The calculation: $n = 1.96^2$ x 0.5(1-0.5) x $1/0.075^2 = 0.9604/0.005625 = 171$. Considering methodological contingencies, the actual sample size for this study will be 188, after adding 10%. Consecutive sampling will be applied to obtain sampling units at the study sites.

**Sample collection and testing.** A data collection tool will be created based on prior studies [30,32]. Sociodemographic and clinical information will be collected from patient charts by direct observation and interviews, supervised by the investigator, two nurses, and two paediatricians over three months. Children's ages (in months) will be categorised as neonates (≤ 1 month), infants (2–12 months), and other children (13–60 months). Clinical data such as comorbidities (for example, HIV, malnutrition, sickle cell disease, pneumonia, anaemia, and congenital anomalies) will be obtained from the medical records. Body weight will be classified as normal (z-score between 2 and −2), underweight (z-score between −2 and −3), or overweight (z-score between 2 and 3), using the WHO Child Growth Standards [47]. Information on previous antibiotic use, prior hospitalisations, intravenous lines, urinary catheters, prematurity, and current antibiotic use will also be collected. Patients will be followed up until discharge, and hospitalisation length and antibiotic use will be recorded.

One to three millilitres of venous blood will be drawn aseptically before antibiotic administration, inoculated into blood culture bottles (BD BACTEC™), and transported to the microbiology laboratory of Ola During Children and Princess Christian Maternity Hospitals daily per standard procedures. Bottles will be incubated at 37˚C for 24 hours. Positive cultures will be sub-cultured on blood, chocolate, and MacConkey agar and incubated at 35˚C to 37˚C for up to 96 hours. Bacterial isolates will be identified using the Vitex 2 compact system or standard biochemical tests [48]. Whole-genome sequencing will be used to detect antibiotic resistance genes using an Illumina genome analyser and sequencer.

Tests will include common first- and second-line antibiotics for BSI. The gram-negative sensitivity card (N291) will include: amikacin, ampicillin, co-amoxiclav, ceftriaxone, ceftazidime, cefepime, cefalotin, cefoxitin, ciprofloxacin, gentamicin, imipenem, meropenem, nitrofurantoin, piperacillin-tazobactam, tigecycline, and trimethoprim/sulfamethoxazole. The gram-positive sensitivity card (GPS67) will include: ampicillin, benzylpenicillin, cefoxitin (MRSA screen), clindamycin, erythromycin, gentamicin, levofloxacin, linezolid, nitrofurantoin, oxacillin, quinupristin, rifampicin, tetracycline, tigecycline, trimethoprim/sulfamethoxazole, and vancomycin. Phenotypes of isolates will be classified according to the breakpoints for MIC, as recommended by the European Committee on Antimicrobial Susceptibility Testing [49]. Participants recruitment and data collection will commence in October 2025 and will be completed by March 2026. The result is expected by June 2026.

**Study variables/outcome variables.** The dependent variables are prevalence of antibiotic-resistant phenotypes of bacterial isolates. The independent variables are age, weight, comorbidities such as HIV infection, malnutrition, sickle cell disease, pneumonia, anaemia, congenital anomalies, previous use of antibiotics before admission, prior hospitalisation, duration, presence of intravenous line, presence of urinary catheter, prematurity, and current antibiotic use.

**Data analysis plan.** Data will be entered into Epicollect5 and analysed using R version 4.3.2. Descriptive statistics will be used to estimate the proportions of children with culture-confirmed BSIs, bacterial species, antibiotic resistance, and other categorical variables. Prevalence of bacterial isolates resistant to at least one antibiotic, multidrug resistance, and extensive drug resistance will also be estimated. Resistance to an antibiotic combination will be assumed if an isolate shows in vitro resistance to each antibiotic. Univariable and multivariable logistic regression will assess the relationship between independent variables and the dependent variables, using odds ratios, 95% confidence intervals, and a 5% significance level.

## Ethical consideration

The Sierra Leone Ethics and Scientific Review Committee (SLESRC) granted ethical approval for this study, with the SLESRC number 019/10/2024. Hospital management will be approached for permission to conduct the research. Healthcare professionals and parents/guardians of children will participate voluntarily with no penalties for non-participation. Written informed consent will be obtained before data collection, and all information will be kept confidential, anonymised, and used exclusively for this study. No identifiable information will be collected in this study.

## Discussion

Given the scarcity of information on antibiotic usage in children under five years of age in Sierra Leone, this research seeks to evaluate the knowledge and perception of paediatric healthcare professionals regarding antibiotic use and AMR. This assessment will highlight areas needing improvement and targeted interventions. Moreover, it is essential to comprehend antibiotic usage patterns in outpatient and inpatient contexts. By examining data using the ATC, DOT/1000PDs, DOT/100BDs, DDD/1000PDs, and DDD/100BDs this study will offer a comprehensive overview of antibiotic use in paediatric populations, identifying trends and potential areas for policy and practice enhancement. Employing these metrics to examine antibiotic consumption provides valuable information regarding antimicrobial use and management initiatives. Additionally, these tools enable standardised evaluations of antibiotic usage across different settings and time frames and allow for comparative assessments between organisations and countries, while facilitating the observation of long-term patterns in antimicrobial consumption [35,50].

Tackling AMR necessitates a comprehensive strategy encompassing improvements in healthcare infrastructure, capacity building, community engagement, and policy formulation. Recognising the gaps in stewardship practices and factors influencing antibiotic use will guide the development of evidence-based interventions and policies to encourage rational use and mitigate the spread of AMR in Sierra Leone.

Regarding some of the study designs that will be employed in this study, self-reported data forms a crucial component of quantitative cross-sectional surveys. While some queries involve direct observation, the majority require participants to remember information and share their personal views, thus introducing the possibility of recall bias. Another limitation is its inability to evaluate incidence and establish causal relationships. Moreover, employing an interviewer-administered response method may induce a response bias, wherein participants feel compelled to provide socially acceptable answers to avoid being perceived as irresponsible. Some of these constraints can be mitigated by utilising validated questions and training research staff to ensure that participants feel at ease.

A summary of data collection tools for all study objectives is included as **Table 1**.

**Table 1. Summary of data collection tools and outcome measures.**

| Data collection tool | Target data source/population | Outcome | Analysis |
|---|---|---|---|
| In-depth interview guide (Form-01) | Paediatric healthcare professionals | Facilitators and barriers to antibiotic prescribing | Deductive and inductive thematic analysis |
| Structured interviewer-based questionnaire (Form-02) | Paediatric healthcare professionals | Knowledge, attitude, and predictors of antibiotic use and resistance | Binary logistic regression and Spearman's rank correlation |
| Record review form-03 | Children under five years | Antibiotic prevalence rate, DOT/1000PDs, DDD/1000PDs and appropriateness | Descriptive statistics, independent t-test, and binary logistic regression |
| Record review form-04 | Children under five years | Antibiotic prevalence rate, DOT/100BDs, DDD/100BDs and appropriateness | Descriptive statistics, independent t-test, and binary logistic regression |
| Record review form-05 | Children under five years | Antibiotic prevalence rate, DOT/100BDs, DDD/100BDs and appropriateness | Descriptive statistics, independent t-test, and binary logistic regression, and ARIMA |
| Record review form 06 | Children under five years | Prevalence of antibiotic-resistant profile, and predictors of resistance | Descriptive statistics and binary logistic regression |

DOT/1,000PDs = Days of therapy/1,000 patient days, DOT/100BDs = Days of therapy/100 bed days, DDD/1,000PDs = Defined daily dose/1,000 patient days, DDD/100BDs = Defined daily dose/100 bed days, ARIMA = Autoregressive integrated moving average.

## Dissemination plan

The findings of this work will be disseminated to healthcare professionals and AMR stakeholders in Sierra Leone, such as the technical working group on AMR and regulatory bodies to inform enforcement strategies and interventions to control irrational antibiotic use. We will also disseminate our findings in academic and professional meetings. Manuscripts will be developed and sent to peer-reviewed journals for publication. In addition, abstracts will be submitted for presentations at local and international scientific and professional meetings. The results of this project will provide evidence-based data to inform the development of policies, guidance, strategies, and interventions to promote antibiotic stewardship in Sierra Leone.

## Author contributions

**Conceptualization:** Onome T. Abiri, Troy D. Moon, John S. Schieffelin, Cynthia A. Adinortey, Gustavo Amorim, Isatta Wurie, Donald S. Grant, Babatunde Duduyemi, Mohamed Samai.

**Methodology:** Onome T. Abiri, Troy D. Moon, John S. Schieffelin, Cynthia A. Adinortey, Gustavo Amorim, Isatta Wurie, Donald S. Grant, Babatunde Duduyemi, Mohamed Samai.

**Resources:** Troy D. Moon, John S. Schieffelin, Isatta Wurie, Donald S. Grant, Babatunde Duduyemi, Mohamed Samai.

**Supervision:** Troy D. Moon, John S. Schieffelin, Cynthia A. Adinortey, Mohamed Samai.

**Writing – original draft:** Onome T. Abiri.

**Writing – review & editing:** Troy D. Moon, John S. Schieffelin, Cynthia A. Adinortey, Gustavo Amorim, Isatta Wurie, Donald S. Grant, Babatunde Duduyemi, Mohamed Samai.

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
