## [Decision Letter · Decision Letter 0]

3 Sep 2025

Dear Dr. Abiri,

Thank you for submitting your manuscript to PLOS ONE. After careful consideration, we feel that it has merit but does not fully meet PLOS ONE’s publication criteria as it currently stands. Therefore, we invite you to submit a revised version of the manuscript that addresses the points raised during the review process.

We look forward to receiving your revised manuscript.

Kind regards,

Usman Rashid Malik

Academic Editor

PLOS ONE

Journal Requirements:

2. Thank you for stating the following financial disclosure: [This research is supported by the Fogarty International Centre of the National Institutes of Health under Award Number D43TW011248. The content is solely the responsibility of the authors and does not necessarily represent the official views of the National Institutes of Health.]. 

Reviewers' comments:

Reviewer's Responses to Questions

**Comments to the Author**

1. Does the manuscript provide a valid rationale for the proposed study, with clearly identified and justified research questions?

Reviewer #1: Yes

2. Is the protocol technically sound and planned in a manner that will lead to a meaningful outcome and allow testing the stated hypotheses?

Reviewer #1: Yes

3. Is the methodology feasible and described in sufficient detail to allow the work to be replicable?

Reviewer #1: Yes

4. Have the authors described where all data underlying the findings will be made available when the study is complete?

Reviewer #1: Yes

5. Is the manuscript presented in an intelligible fashion and written in standard English?

Reviewer #1: Yes

You may also provide optional suggestions and comments to authors that they might find helpful in planning their study.

Reviewer #1: The article is well written and addresses a very current and important issue. In addition, the study uses a mixed (qualitative and quantitative) approach to analyze knowledge, perceptions, and antibiotic prescribing practices, providing a comprehensive view of the problem. The objectives were clearly defined, and the use of standardized metrics allows for comparable and reliable evaluation of the data. However, the study could have biases, for example, the use of questionnaires, long timelines, limited resources, and narrow focus as a population group. Although the authors included multiple centers from different parts of the country in the study, this is positive because it allows for a heterogeneous population but also limiting because it may not fully reflect the reality of the country.

**Do you want your identity to be public for this peer review?** For information about this choice, including consent withdrawal, please see our Privacy Policy

Reviewer #1: No

---

## [Author Response · Author response to Decision Letter 1]

16 Sep 2025

Prof Usman Rashid Malik

Academic Editor

PLOS ONE

Dear Sir,

Response to editorial and reviewers comments

Authors response: The manuscript has been formatted according to the guidelines above and the evidence are in track changes.

2. Thank you for stating the following financial disclosure: [This research is supported by the Fogarty International Centre of the National Institutes of Health under Award Number D43TW011248. The content is solely the responsibility of the authors and does not necessarily represent the official views of the National Institutes of Health.].

Authors response: We have added ""The funders had no role in study design, data collection and analysis, decision to publish, or preparation of the manuscript"" in the manuscript and the cover letter.

Authors response: In addition to what is already captured under data availability, from 528-536, we included a whole section on dissemination plan or data sharing plan which reads - Dissemination plan

The findings of this work will be disseminated to healthcare professionals and AMR stakeholders in Sierra Leone, such as the technical working group on AMR and regulatory bodies to inform enforcement strategies and interventions to control irrational antibiotic use. We will also disseminate our findings in academic and professional meetings. Manuscripts will be developed and sent to peer-reviewed journals for publication. In addition, abstracts will be submitted for presentations at local and international scientific and professional meetings. The results of this project will provide evidence-based data to inform the development of policies, guidance, strategies, and interventions to promote antibiotic stewardship in Sierra Leone.

Authors response: We have moved it to the materials and methods section.

Authors response: The map was made using ArcGIS and did not include any proprietary software and as such has no copyright associated with it.

Authors response: There are no recommendations regarding citations.

Authors response: We have not cited any retracted article.

Yours Sincerely,

Onome T. Abiri

---

## [Editor Report · Decision Letter 1]

3 Nov 2025

A study protocol for the assessment of antibiotic use and bacterial antimicrobial resistance among children under five years of age: implications for a resource-limited setting

PONE-D-25-38721R1

Dear Dr. Abiri,

We’re pleased to inform you that your manuscript has been judged scientifically suitable for publication and will be formally accepted for publication once it meets all outstanding technical requirements.

Kind regards,

Usman Rashid Malik

Academic Editor

PLOS ONE
---

## [Editor Report · Acceptance letter]

PONE-D-25-38721R1

PLOS ONE

Dear Dr. Abiri,

I'm pleased to inform you that your manuscript has been deemed suitable for publication in PLOS ONE. Congratulations! Your manuscript is now being handed over to our production team.

Kind regards,

on behalf of

Dr. Usman Rashid Malik

Academic Editor

PLOS ONE